# Structurally Different Yet Functionally Similar: Aptamers Specific for the Ebola Virus Soluble Glycoprotein and GP1,2 and Their Application in Electrochemical Sensing

**DOI:** 10.3390/ijms24054627

**Published:** 2023-02-27

**Authors:** Soma Banerjee, Mahsa Askary Hemmat, Shambhavi Shubham, Agnivo Gosai, Sivaranjani Devarakonda, Nianyu Jiang, Charith Geekiyanage, Jacob A. Dillard, Wendy Maury, Pranav Shrotriya, Monica H. Lamm, Marit Nilsen-Hamilton

**Affiliations:** 1Ames Laboratory, U.S. Department of Energy, Ames, IA 50011, USA; 2Roy J. Carver Department of Biochemistry, Biophysics and Molecular Biology, Iowa State University, Ames, IA 50011, USA; 3Department of Mechanical Engineering, Iowa State University, Ames, IA 50011, USA; 4Aptalogic Inc., Ames, IA 50014, USA; 5Department of Microbiology and Immunology, University of Iowa, Iowa City, IA 50011, USA; 6Department of Chemical and Biological Engineering, Iowa State University, Ames, IA 50011, USA

**Keywords:** aptamers, Ebola GP1,2, Ebola sGP, electrochemical sensor, aptamer–protein interaction

## Abstract

The Ebola virus glycoprotein (GP) gene templates several mRNAs that produce either the virion-associated transmembrane protein or one of two secreted glycoproteins. Soluble glycoprotein (sGP) is the predominant product. GP1 and sGP share an amino terminal sequence of 295 amino acids but differ in quaternary structure, with GP1 being a heterohexamer with GP2 and sGP a homodimer. Two structurally different DNA aptamers were selected against sGP that also bound GP1,2. These DNA aptamers were compared with a 2′FY-RNA aptamer for their interactions with the Ebola GP gene products. The three aptamers have almost identical binding isotherms for sGP and GP1,2 in solution and on the virion. They demonstrated high affinity and selectivity for sGP and GP1,2. Furthermore, one aptamer, used as a sensing element in an electrochemical format, detected GP1,2 on pseudotyped virions and sGP with high sensitivity in the presence of serum, including from an Ebola-virus-infected monkey. Our results suggest that the aptamers interact with sGP across the interface between the monomers, which is different from the sites on the protein bound by most antibodies. The remarkable similarity in functional features of three structurally distinct aptamers suggests that aptamers, like antibodies, have preferred binding sites on proteins.

## 1. Introduction

Viral outbreaks are a growing concern with serious socio-economic impacts [1,2]. Rapid isolation of contagious individuals is critical to stemming the progress of diseases such as Ebola virus disease, which causes severe hemorrhagic fever in humans and non-human primates. Its fatality rate can reach 90% and averages around 50% [3,4,5]. With early detection, infected individuals can be isolated and access treatment, which prevents the spread of the virus through a susceptible population.

The *Zaire Ebolavirus* (EBOV) glycoprotein gene produces three protein products, which are the surface glycoprotein (GP1,2) and two secreted proteins (sGP and ssGP) [6,7]. The Ebola viral soluble glycoprotein (sGP) is translated from 73% of the transcripts of the GP1,2 gene [8] and can be detected in significant amounts in the sera of acutely infected patients within 3 days of exposure to the virus [9,10].

Motivated by the need for sensitive biosensors to detect *Ebolavirus* infections early during infection, we isolated single-stranded DNA (ssDNA) aptamers that recognize sGP and compared the properties of these aptamers with a previously isolated 2′FY-RNA aptamer [11]. Apart from the DNA aptamers reported in this work and the previously reported 2′FY-RNA aptamer [11], we are unaware of other aptamers reported to recognize Ebola virus sGP. However, DNA aptamers have been reported to have high affinity for the EBOV surface glycoprotein [12]. It is unknown if these aptamers recognize sGP. The ability to recognize sGP is important for early identification of *Ebolavirus* infections, as this protein is expressed at a molar ratio of ~3:1 compared to GP1,2, is secreted into the bloodstream and has an effective concentration for detection that is much higher than for GP1,2, which is bundled on viral particles.

The Ebola virus spreads by direct contact with broken skin or mucous membranes. The incubation period between exposure to the onset of illness with viremia is typically 3 to 13 days but may be as long as 21 days [13,14]. It is critical to develop sensors for early detection before viremia is evident, and the infected individual is highly contagious. With current sensors, it can take 3 days after the symptoms start for the virus to be detectable [3]. The WHO recommends nucleic acid tests (NAT) and rapid antigen detection tests for use in remote settings where NATs are not readily available [14]. The available rapid tests provide qualitative identification of Ebola antigens and are mostly recommended for screening purposes to be confirmed with NATs. Detectable sGP levels in the blood of non-human primates correlate with virus replication dynamics [15]. Thus, a sensor that recognizes sGP and can be readily deployed on-site even in the most remote parts of the world would provide a valuable addition to analytical options to prevent viral spread.

Advances in nanotechnology and micro-electronics, coupled with the revolutions in big data and wireless communications, make it possible to develop portable and low-cost electrochemical biosensors. The use of impedance spectroscopy-based devices requires only the skill of operating a smartphone and can produce results in less than 1 h [16,17]. The main requirement for a well-functioning device is the incorporation of a robust recognition element and a transducer interface that can facilitate the sensitive discrimination of a target in a complex matrix. In the present work, a prototype for a DNA-aptamer-based, low-cost POC diagnostic device is reported, which can detect an Ebola biomarker with high fidelity. This electrochemical aptasensor, with its simple design and nonreliance on electrical appliances or cooling devices, could be developed for employment in remote areas where *Ebolavirus* outbreaks often begin.

Although our target for aptamer selection was sGP, the 295 amino acid identity in primary structure between sGP and GP1 suggests the possibility that aptamers might bind both proteins. However, despite their N-terminal sequence and similarities in domain structures [18], sGP and GP1 have different three-dimensional structures. sGP forms a soluble homodimer, and GP1 forms a transmembrane heterohexamer with GP2. The difference in quaternary structure between sGP and GP1,2 was expected to be sufficient to allow discrimination between these proteins by aptamers, as observed for most antibodies. Conformational monoclonal antibodies show strong biases for binding either sGP or GP1,2 [19]. Unexpectedly, we found that the three aptamers, which are not similar in primary or predicted secondary structures and are highly specific for sGP over many other serum proteins, also bind GP1,2 with similar affinities. We explored this observation experimentally and computationally. Our findings suggest that the aptamers bind across the monomer interface on the sGP dimer. All three aptamers compete for binding to sGP and GP1,2, which we interpret to mean that they bind to equivalent sites on sGP and GP1,2. Computational modeling also suggests that these aptamers interact with the same region across the monomer interface on sGP. Experimental and modeling results show that this region is not the same as bound by three monoclonal antibodies raised to GP1,2 and one polyclonal antiserum raised to a peptide common to sGP and GP1.

Overall, our findings showed remarkable similarity in the binding isotherms and sites of interaction of three structurally different aptamers that bind two structurally different but related proteins with high affinities. The results suggest that, like for antibodies, aptamers have preferences for binding certain features on proteins that result in affinities equivalent to antibodies for the same proteins.

## 2. Results

### 2.1. Aptamer Selection

DNA aptamers with high affinities for EBOV sGP were selected from an oligonucleotide pool consisting of a central 53 nt random sequence region flanked by 25 bases and 5′ and 3′ constant regions with a starting complexity of ~10^15^ molecules. The SELEX protocol followed a mathematically defined approach of starting with a relatively high molar ratio of 2:1 (ssDNA: sGP), followed by harmonic reductions in the EBOV sGP concentration [20] to increase the stringency of selections (Figure 1A). Oligonucleotides with nonspecific binding to the nitrocellulose membrane and/or human serum albumin (HSA) were discarded with a counter-selection after round 4, which involved one round of incubating the oligonucleotide pool with has, followed by passage through the nitrocellulose filter, but no PCR amplification prior to subsequent additional rounds of positive selection. After each round of selection, the bound oligonucleotides were extracted and quantified by their absorbance at 260 nm. The increase in the percentage of sGP bound by ssDNA from the pool (Figure 1B) is an indication of enrichment of the sGP binding oligonucleotides in the ssDNA pool during selection. After eight selection rounds (Figure 1A), the remaining oligonucleotide pool was cloned, and 53 oligonucleotide sequences were obtained (Figure 1C). Oligonucleotides representing the major clades were tested for binding by the dot-blot assay. Oligonucleotides 6011 and 6012 (Figure 1) were identified as high-affinity binding aptamers.

### 2.2. Aptamer Affinities and Specificities

Despite their divergent sequences and no identifiable sequence motif, oligonucleotides 6011 and 6012 bound EBOV sGP with similar high affinities, and also bound Sudan virus (SUDV) sGP, but with slightly lower affinities (Figure 2A,B, Table 1). Oligonucleotide 6011 did not bind another histidine-tagged protein (recombinant Lcn2), nor HSA (Figure 2). Binding to both SUDV and EBOV sGPs, which are the most evolutionarily divergent of all the Ebola virus species, suggests that the oligonucleotides bind to a conserved motif in sGP. The affinities of 6011 and 6012 were compared with the affinities of a previously selected 2′FY-RNA aptamer (39SGP1A) for binding EBOV sGP and GP1,2 (Table 1).

The three aptamers were also tested for their abilities to bind GP1,2. They all bound to GP1,2 with similar affinities as for sGP (Figure 2C, Table 1). These affinities for sGP and GP1,2 are in the range of those measured for several neutralizing monoclonal antibodies [21], including KZ52 [22]. However, antibodies generally show strong preferences for binding one protein or the other [19,21]. This unexpected finding for the selected oligonucleotides suggests that they may be binding a region of shared structure between sGP and GP1,2 [18]. As GP1, and not GP2, shares an amino acid sequence and some structural features with sGP, we presume that the aptamers bind GP1 and not GP2.

### 2.3. Truncation to Produce Aptamer 70SGP2A

Oligonucleotides isolated by SELEX are all the same length as in the original library, which was 100 nt in this study, and the selected oligonucleotides often contain an embedded aptamer sequence. Being embedded in a longer sequence can alter the properties of the aptamer and give it undesirable features, as we observed for 39SGP1A, which does not bind albumin but was embedded in an oligonucleotide sequence that bound albumin [11]. Analysis of the abilities of truncated versions of 6011 to bind sGP, the predicted secondary and tertiary structures, and results from computationally docking folded 6011 to sGP, combined to suggest that the 3′ 30 bases of 6011 did not contribute to binding of sGP. The aptamer 70SGP2A was created from the first 70 nt of 6011. 70SGP2A bound sGP and GP1,2 with similar affinities to that of 6011 (Figure 3C, Table 1). Further truncation of 6011 to a 53-mer (70SGP2ATR5′53) resulted in no binding to sGP (Figure 3A,B). 70SGP2A binding activity was specific to sGP and GP1,2 and did not bind a series of serum proteins (Figure 3D). Similar results were found for 39SGP1A [11] and oligonucleotide 6012 (Appendix A). That these aptamers did not bind to albumin, α1-antitrypsin, α2-macroglobulin, IgG, rheumatoid factor, or fibrinogen, which together account for more than 75% of the protein in serum, suggested that they might be developed into accurate sensing elements for analyzing GP1,2 and sGP concentrations in serum.

### 2.4. Protein Surface Epitope for Aptamer Binding

To identify the epitope on sGP bound by each of the aptamers, we first investigated if all aptamers bind to the same location on sGP. If so, the aptamers would compete for binding to sGP, as was observed (Figure 4A). We then asked if the aptamer binding site on sGP overlaps with the antibody binding sites by determining if antibodies compete with aptamer binding. In preliminary studies, we tested three monoclonal antibodies that bind to different epitopes on GP1,2 and sGP for their abilities to compete with the aptamers for sGP binding [21,23]. We also tested a rabbit polyclonal antiserum raised against residues 83–96 (TKRWGFRSGVPPKV) of pro-sGP that are present in both sGP and GP1. The EC50 for sGP was determined for each of the immunological reagents used in the competition studies (Appendix A). A series of 2′FY-RNA and DNA oligonucleotides that bind EBOV sGP were tested for competition with these antibodies for sGP binding in an ELISA (Appendix A). No competition was observed. Using radiolabeled aptamers, we tested the ability of radiolabeled oligonucleotides 6011 and 6012 to compete with the monoclonal antibodies FVM-04 and FVM-09, for binding to sGP and found that these antibodies do not compete (Figure 4B). Antibody FVM-04 binds a conformational epitope within the core of GP1 (residues 31 to 200), and FVM-09 binds EBOV GP residues 286–290 (GEWAF) [21]. Both antibodies also bind sGP (Appendix A). The affinities of the aptamers, 6011 and 6012, for sGP are in the range of those reported for a series of antibodies that recognize sGP [21,22]. Therefore, a 6–10-fold excess of each oligonucleotide (Appendix A and Figure 4B) is expected to displace the antibodies present at their EC50 concentrations.

We investigated the impact of the quaternary structure of sGP on aptamer binding and asked if the dimer was necessary for aptamer binding. Binding by oligonucleotide 6011 is disrupted when sGP is incubated with the reducing agent, dithiothreitol (DTT), to dissociate the dimer (Figure 4C). Similar results were previously found for 39SGP1A [11]. These results suggest that the aptamers bind to a single site on sGP that is different from where the antibodies bind, and that the site is disrupted when the protein is reduced and dissociated into monomers. Another explanation for the decrease in aptamer binding to sGP might be a loosening of the sGP monomer structure due to the reduction of its intramolecular disulfide bond. However, this explanation is unlikely because the same experiment performed with GP1,2 with 6011 or NGAL with its aptamer (NA53) [24] showed no effect of DTT on aptamer binding (Figure 4D). GP1,2, sGP and NGAL have three, one, and one internal cystines, respectively.

### 2.5. Molecular Modeling to Explore the Probable Aptamer Docking Site on sGP

To explore the likely interactions between the aptamers and sGP, we created three dimensional models of these aptamers (Appendix A) and docked them to sGP. Most resulting dockings showed the aptamer binding to the region over the interface between the monomers of sGP (Figure 5A,B). This region on sGP for aptamer–protein interaction was seen with 6 of the 10 docking models for 70SGP2A and sGP (Appendix A).

The experimental results showed that all aptamers bound both EBOV and SUDV sGP, which suggested that there should be some conserved surface structure in the region of aptamer binding. A comparison of the EBOV and SUDV sGP structure with surface residues colored to indicate conservation (white) or divergence (red) shows that the region of the interface contains both conserved and divergent amino acid residues (Figure 5C). The residues along the interface are also a mixture of basic and acidic residues (Figure 5D), and the docked aptamers were found to interact with residues of both charges.

In contrast to the proposed aptamer binding site, the antibodies tested bind to different regions on sGP (Figure 5G and Appendix A). The distribution of conserved and divergent amino acid residues in the region proposed to be bound by the aptamers is consistent with the lower affinity for SUDV sGP than for EBOV sGP by aptamers that were selected against the latter.

The surprising observation that the three aptamers bind GP1,2 with similar affinities cannot be explained easily by there being a similar exposed interface on GP1,2 as on sGP. The interface residues identified on sGP are largely buried in the GP1 monomer, leaving only a few residues exposed (Figure 5E,F). This result suggests that either the common binding interface of the aptamers to sGP and GP1,2 does not involve interface residues but instead is constituted of nearby residues with similar surface configurations on the two proteins, or that the structures of these proteins are pliable in solution and expose structurally similar sites for aptamer binding that are not evident when these proteins are packed in the crystal lattice necessary for collecting data by X-ray crystallography.

### 2.6. Aptamer Structural Stabilities in the Presence of Serum

To evaluate the suitability of the aptamers as molecular recognition elements in a sensor for early diagnosis of *Ebolavirus* infection in patient serum, we tested their structural stabilities in presence of serum. Oligonucleotides 6011, 6012, and 39SGP1A were stable over 24 h at temperatures ranging from 24 to 58 °C in the presence of serum (Figure 6A and Appendix A).

The long-term stability of the aptamers in the presence of 50% serum was tested in the presence and absence of 1 mM EDTA. Whereas 6011 had a half-life of about 12 days in the absence of EDTA and 32 days in its presence (Figure 6B), the 2′FY-RNA aptamer, 39SGP1A, was stable for much longer. The main contribution to the loss of 6011 with time seems to be due to components of serum, as 6011 was perfectly stable for over a month at 24 °C in the absence of serum (Figure 6B). As expected, 39SGP1A, being a 2′FY-RNA aptamer, was protected from nuclease action. A DNA oligonucleotide of the 39SGP1A sequence had similar stability to 6011.

These results show that both DNA and 2′FY-RNA aptamers are sufficiently stable in the presence of serum to be used as detection elements for sensors for analyzing blood samples but that a 2′FY backbone greatly extends the oligonucleotides’ lifetime.

### 2.7. Electrochemical Assay and Detection of sGP and GP1,2

Oligonucleotide 6011 was incorporated into a nanoporous anodic aluminum oxide (NAAO) electrochemical sensor [25] and used to quantify sGP in the presence or absence of 10% human serum. The 6011-aptasensor (Figure 7A) demonstrated a Kd of 2.2 ± 0.75 nM with a linear range from 0.15 to at least 44 nM, which was the highest concentration tested, and a limit of detection (LOD) of 150 pM EBOV sGP (Figure 7B and Appendix A). The selectivity of the system was tested in 10% human serum for sGP with a 92% confidence interval. The sensor did not detect HSA (Figure 7B), and membranes not conjugated with aptamers gave no signal with sGP (Figure 7B). The same affinity was found for SUDV sGP as for EBOV sGP.

We also tested serum from a macaque monkey infected with EBOV for sGP content (Figure 7C). The amount of sGP measured by the NAAO sensor was 1 to 1.3 µM, which compares well with 0.5–3.5 µM determined by Western blot analysis, the 0.65 + 0.3 µM determined by ELISA for this sample, the 1 µM estimated from literature reports for Guinea pig samples based on quantification of shed GP1,2, and the knowledge that the relative rate of sGP:GP1,2 production is ~3:1 [6,26].

The 6011-aptasensor was also tested for its ability to recognize Ebola GP1,2 pseudotyped VSV virions in comparison with control vesicular stomatitis virions (VSV) that do not display GP1,2 on their surfaces (Figure 7D). The results showed that the 6011-aptasensor can detect the Ebola virus GP1,2 VSV pseudovirion particles at low densities. We also tested two other oligonucleotides isolated from the same library as 6011 and 6012 for their functions on the sensor. Oligonucleotide 6020 does not bind sGP in solution (Appendix A), nor does it bind GP1,2 pseudotyped virions on the sensor (Figure 7D). Oligonucleotide 6022 binds sGP in solution (Appendix A) and binds GP1,2 pseudotyped virions on the sensor (Figure 7D). It should be noted that pseudotyped virion preparations from cell cultures also contain exosomes with a similar membrane composition as the virions. Exosomes are larger than virions and would not be counted by DLS as virions; however, they likely have GP1,2 on their surfaces. Thus, we are unable to determine the LOD for virion detection due to the high probability that the sensor detects virions and exosomes. The latter are present at higher densities than the virions in the five preparations that we examined by transmission electron microscopy. Thus, our finding that the sensor detects GP1,2 on pseudotyped virions should be interpreted as the sensor detecting GP1,2 in the context of a membrane such as that which encases virions. Quantification and determination of LOD for virions will require a homogeneous virion preparation that lacks cellular components, which is currently unavailable.

## 3. Discussion

### 3.1. Aptamer Characteristics

Our current study identified two DNA aptamers that differ in sequence and predicted structure but that bind sGP with similar high affinities and specificities. Here, we have compared the properties of these DNA oligonucleotides and the truncated version of 6011 (70SGP2A) with a previously isolated 2′FY-RNA aptamer (39SGP1A). Despite the differences in length, predicted secondary structure, and backbone chemistry, these oligonucleotides displayed similar high affinities for sGP and GP1,2 while not binding a series of other serum proteins.

*Ebolavirus* sGPs are more conserved than the viral surface glycoproteins [9], and Ebola virus species EBOV and SUDV are the most divergent Ebola species [27]. Their mature sGP sequences are 68% identical, having considerable variation in surface-exposed side chains. Despite this variation, both DNA oligonucleotides 6011 and 6012 and 70SGP2A bound sGP from both species, as did the 2′FY-RNA aptamer 39SGP1A (This work and [11]). As expected, their affinities for EBOV sGP (the selection target) were higher than for SUDV sGP, but as for EBOV sGP, the affinities of all three aptamers for SUDV sGP were in the same numerical range.

That all three aptamers bound sGP and GP1,2 with similar affinities despite the fact that the two proteins share their first 265 amino acid residues was surprising. This is because sGP and GP1,2 adopt different tertiary and quaternary structures, although a close comparison between the structures of these two proteins showed some similarities in the crystal lattice [18]. These protein structures may also be more pliant in solution. For example, when co-expressed with GP2, sGP can substitute for GP1 in creating infectious virions [28]. The GP1,2 heterohexamer is recognized by the antibody KZ52, which binds to the base of the complex with an epitope that encompasses GP1 and GP2 [18]. KZ52 was also equally effective at neutralizing the infectivity of virions co-transfected with GP1 and GP2 as with cells co-transfected with sGP and GP2 [28]. Thus, although sGP and GP1 structures in crystals are different, it appears that they can adopt similar structures on the virion. These proteins may be capable of adapting to alternate structures in solution such as might happen if the interaction between aptamer and protein is a process of mutual induced fit.

To establish that the aptamers were sufficiently resistant to nucleases in serum to be used in a sensor for blood proteins, we evaluated the half-lives of the DNA and 2′FY-RNA aptamers by determining their rates of loss. The 2′FY-RNA aptamer, 39SGP1A, was stable over the 80-day period of the assay. However, the 39SGP1A sequence of DNA demonstrated the same stability as the DNA oligonucleotide 6011. Thus, the reason for the very high stability of 39SGP1A in serum is the presence of the 2′ pyrimidine modification [29,30]. We also found that the melting temperature of 39SGP1A was higher and much more homogeneous than that of the RNA version of this aptamer (Appendix A), a result which is consistent with the observation that 2′FY-RNA shows increased Watson–Crick bonding strength relative to RNA [31].

### 3.2. Application of Aptamers as Recognition Elements on an Electrochemical Sensor

Nanoporous anodized aluminum oxide membranes (NAAO) have several desirable properties, which make them attractive for use in a variety of biosensing platforms. Some of the salient properties are non-conductivity, well-defined nanopores, small pore size, high pore density, and ease of functionalization. Here, we built on previous successful applications of NAAO membranes in electrochemical sensors [32,33,34] to develop an aptasensor that recognizes sGP and GP1,2. In our sensor design, the change in impedance observed upon ligand binding has been attributed to charge modulation in the asymmetric gold-coated nanopores [35].

A DNA aptamer (6011) was chosen for initial tests of the applicability of aptamers for detecting sGP and pseudovirions bearing EBOV GP in serum-containing samples. The aptamer selectively detected sGP and GP1,2 in solution and GP1,2 on the virion when incorporated into a nanoporous aluminum oxide (NAAO)-based electrochemical sensor. A second aptamer from the same selection as 6011 and 6012 also showed a similar binding isotherm to the pseudotyped virion. An oligonucleotide that does not bind sGP was tested and gave no signal, unlike an NAAO membrane that was not functionalized with an aptamer. These results show that the aptamer-functionalized NAAO membrane sensor could specifically recognize sGP and GP1,2 by a mechanism that reflected the aptamer’s binding capability and that required the aptamer’s presence.

We note that the calculated K_d_ of the aptasensor for sGP was 2.2 nM, whereas the Kd determined for oligonucleotide 6011 in solution was 8.5 nM (Table 1). We have previously observed that the K_d_ measured with an aptamer linked to surface is lower than the K_d_ measured in solution [36]. One explanation for this observation might be molecular crowding, which promotes nucleic acid folding, and which occurs when aptamers are confined to a surface [37,38,39].

The sensor also recognized sGP in a sample of serum from an EBOV-infected monkey and delivered a quantitative result that agrees with our ELISA and Western blot analysis and with the reported concentrations of sGP in the sera of infected individuals. Thus, this sensor is a prototype for a hand-held device to identify individuals early in *Ebolavirus* infection and has the advantage over antibody-powered sensors of recognizing sGP and GP1,2 from at least two *Ebolavirus* species. Our sensor compares favorably with other reported sensors for *Ebolavirus* (Appendix A), particularly because it can detect sGP, which is found in the blood early after infection and is produced in a molar ratio to GP1,2 of ~3:1. The sensor can produce a qualitative (YES/NO) result in 30 min, whereas a quantitative determination can be performed in 3 h (Appendix A).

### 3.3. Protein Site(s) to Which the Aptamers Bind

Antibody affinities for sGP and GP1,2 have been reported to be in the range of 4 to 200 nM, depending on the antibody [18,22]. As these affinities are in the range of the affinities determined for the aptamers in this study, we investigated the overlap in protein epitopes recognized by 6011, 6012, and 39SGP1A with three monoclonal antibodies and an antiserum. These reagents were chosen because they have critical contact residues within the 295 amino acid stretch of identity between sGP and GP1,2. For FVM-04 and FVM-09, EBOV GP residues K115, D117, and G118 are critical for binding [40]; and W288, F290, and W291 are essential for FVM-09 binding to GP1,2 [41]. Antibody 21D10 binds residues 81–90 (SATKRWGFRS) in the full-length protein, which correspond to residues 29–38 on the PDB structure (Figure 5G). None of the aptamers competed for binding with the antibodies, but all the aptamers competed with other aptamers for binding sGP. This observation suggests that proteins may have a limited number of sites that are favorable for aptamer binding, which we will refer to as aptatropic sites to distinguish them from the epitopes recognized by antibodies. Our analysis suggests that sGP has one aptatropic site. Further evaluation of this hypothesis will require the identification of the binding location(s) of other independently isolated aptamers with different structures, such as those selected against GP1,2 [12].

The observation that dithiothreitol, which dissociates the sGP dimer, destroys binding by 6011 and 6012, suggests that the aptamers bind over the interface between the monomers in the dimer. We previously reported a similar observation for 39SGP1A [11]. Control experiments showing no effect of DTT on the interaction of oligonucleotide 6011 with GP1,2 or an NGAL aptamer with NGAL suggest that the reason for the effect of DTT on sGP binding is due to dissociation of the sGP dimer rather than distortion of the protein tertiary structure due to reduction of internal cystines. Aptamer binding to sGP, predicted by computational analysis with docking, also identified a region surrounding the dimer interface as the most likely location for aptamer binding to sGP.

### 3.4. Distinction of Antibody and Aptamer Binding Sites on sGP

Conformational epitopes for antibodies have been well-studied, and compared with other heterodimeric protein interfaces, they have more options for H-bond, cation–π, amino–π, and π–π interactions; more charged and aromatic residues; and fewer aliphatic residues with less hydrophobic packing [42]. Many of the characteristics described for antibodies are consistent with our ideas of where aptamers should bind. Therefore, we were surprised to observe that aptamers compete with other aptamers, but not with antibodies, for binding sGP. Aptatropic sites on proteins have not been well defined or even recognized to exist. However, our observation for sGP is consistent with many reports of assays that require the simultaneous binding of antibodies and aptamers to the same protein, which can only occur if the aptamers and antibodies bind to different sites on each protein [43,44,45,46,47].

That aptamers can identify binding sites on proteins that are not bound by antibodies might make them excellent sensing elements to detect viruses that are selected in populations due to pressure of neutralizing antibodies. Aptasensors might be insensitive to mutants of viral glycoproteins that evolve to avoid antibody capture. In that regard, it will be of interest to compare the rates of evolution of antibody epitopes with aptatropic sites on virus proteins once these latter sites are identified.

Antibody binding epitopes have several characteristics that might distinguish them from nucleic acid binding epitopes, such as that they are small and more frequently contain loops than helices. These features define the antibody binding epitope as a flexible region in the target protein and have led to the “flexible lock–adjustable key” model for antigen–antibody interactions, in which the conformational mobility of antibody binding sites is balanced by a similar disorder in the target epitope [48]. By comparison to proteins, aptamers are extremely flexible molecules. Thus, the balance of flexibility and rigidity for aptamer–protein interactions may be quite different than for protein–protein interactions. Electrostatic interactions contribute to binding for antibodies and aptamers [49,50], and sites on proteins that are positively charged can be predicted to attract nucleic acids. The proposed binding site on sGP has a strip of basic residues, but electrostatics alone are not sufficient to explain the high affinities and specific interactions observed for the aptamers described here. This is evident from the modeled aptamer–protein complexes (Figure 5A,B), which show that the contacts with the protein do not follow the strip of basic amino acids. The analysis of one model for interacting residues revealed basic, acidic, and neutral amino acids. In addition to electrostatic bonds, other noncovalent bonds, including H-bonds, cation–π, amino–π, and π–π interactions are involved in establishing the tight selective fits of aptamers to proteins [51,52,53,54,55,56].

### 3.5. Summary

In summary, we have compared the properties of three aptamers isolated to bind the target protein EBOV sGP. These aptamers differ in sequence and structure, yet bind with remarkably similar relative affinities to sGP from EBOV and SUDV and EBOV GP1.2. These aptamers are highly specific for the GP gene products. One aptamer was demonstrated as an effective recognition element in an electrochemical sensor to detect sGP and GP1,2 in solution and GP1,2 in the context of a membrane.

Our results point to the possibility that specific regions on the surfaces of proteins might be aptatropic. Identifying the critical features of such sites, along with improved 3-D structural predictions for aptamers, would increase the accuracy of predicting aptamer interaction sites on proteins, identify proteins with a high likelihood for high affinity aptamer binding, and identify which protein mutation(s) might significantly alter aptamer binding. Such a detailed structure-based understanding of aptamer–protein interaction is also needed to improve the computational predictability of aptamer protein binding, and perhaps eventually to obviate the need for the experimental SELEX protocol.

## 4. Materials and Methods

### 4.1. Buffers

Buffers used in this study were: BW (2.0 M NaCl, 1mM EDTA, 10 mM Tris·HCl, pH 7.5), PS (137 mM NaCl, 2.7 mM KCl, 10 mM Na_2_HPO_4_, 2 mM KH_2_PO_4_, pH 7), PBS (137 mM NaCl, 2.7 mM KCl, 10 mM Na_2_HPO_4_, 2 mM KH_2_PO_4_, 5 mM MgCl_2_, pH 7), PS-T (PS, 0.05% Triton X-100), NBB (50 mM Tris-HCl, 150 mM NaCl, 5 mM KCl, 1 mM MgCl_2_, pH 7.4), TAE (40 mM Tris acetate, 1 mM EDTA, pH 8), BSM1C (5 mM Na_2_HPO_4_, 5 mM KH_2_PO_4_, 1 mM MgCl_2_, 3 mM NaN_3_, pH 7), RSBA (50 mM Tris-HCl, 150 mM NaCl, 5 mM KCl, 1 mM MgCl_2_, 10 mM NaN_3_, pH 7.4), TBE (90 mM Tris-borate, 2 mM EDTA, pH 8).

### 4.2. Ebola Virus Proteins

Recombinant Sudan virus soluble glycoprotein (SUDV sGP) was purchased from IBT Bioservices (Rockville, MD, USA, Cat. No. 0570-001). Recombinant EBOV GP1,2 was purchased from BPS bioscience (San Diego, CA, USA, Cat. No. 21003) and Sino Biological (Wayne, PA, USA, Cat. No. 40459-V08H). The sequence of the recombinant EBOV GP1,2 is compared with the sequence of GP1 in the pdb dataset 5KEN in the Appendix A.

Murine Lcn2 and human NGAL were produced from transformed *E. coli* using TALON Metal Affinity Resin (BD Biosciences) and eluted with imidazole. EBOV sGP protein was produced in 15 cm plates of HEK 293T cells by polyethylenimine (PEI) transfection with 30 µg of the expression vector [57]. The next day, media was replaced with Opti-MEM low-serum media containing 1 mM HEPES and 1% non-essential amino acids (ThermoFisher/Gibco, Waltam, MA, USA, Cat. No. 31985062). Supernatant was collected at 48 and 72 h following transfection, filtered through a 0.45 µ filter, and frozen at −80 °C until purified. Supernatants were thawed and concentrated by centrifugation using an Amicon Ultra-15 centrifugal filter units (Millipore Sigma, Burlinton, MA, USA, Cat. No. C7715) with 10 kDa molecular weight cut-off (MWCO). sGP-6xHis present in the concentrated supernatants was bound to and eluted from cobalt-bound Dynabeads using the Dynabead His-Tag Isolation and Pulldown Kit (Dynal Biotech, Bengalore, India/ThermoFisher, Cat. No. 10103D) protocol. Eluted samples were passed through Zeba Spin Desalting Columns (ThermoFisher, Cat. No. 89882). The desalted material was washed 5–10 times with 400 µL of DPBS in an Amicon Ultra-0.5mL centrifugal filter unit as described above. Final purified samples were diluted to desired concentration in DPBS. Purified sGP was analyzed by Coomassie Blue staining and Western blotting using anti-pan filovirus GP monoclonal antibody, 21D10 (IBT Bioservices). The sequence of the purified sGP is compared with the sequence of sGP in the PDB dataset 5KEM in the Appendix A.

### 4.3. Aptamers and Oligonucleotides

All oligonucleotides used in this study were purchased from IDT (Coralville IA). Most purchases were for oligonucleotides prepared by the standard desalting procedure. Oligonucleotides were evaluated by acrylamide gel electrophoresis to establish that they were homogeneous and of the expected molecular weight. Oligonucleotide sequences are shown in Table 2.

### 4.4. Ebola Pseudovirions

Preparations of EBOV pseudovirions were performed as previously described [57,58]. Briefly, HEK 293T cells were transfected with an EBOV GP-expressing construct using polyethylenimine (PEI) transfection reagent. Sixteen to twenty-four hours later, transfected cells were infected with a VSV pseudovirus stock bearing Lassa virus GP (MOI = ~1). These pseudovirions contain the VSV genome, which lacks the native G glycoprotein gene. Six hours later, the cells were washed thoroughly to remove input virus, and supernatant-containing VSV pseudovirions bearing EBOV GP were collected at ~36 and 48 h following initial transfection, filtered through a 0.45 μm filter, and frozen until purified. For large preparations of virus stocks, supernatants were thawed and centrifuged at 7000× *g* overnight at 4 °C. The resulting pellet was resuspended in PS and layered over a 20% or 25% sucrose cushion and ultracentrifuged at ~82,000× *g* for 2 h. The pellet was resuspended in PS, aliquoted, and stored at −80 °C until use. To determine the density of infectious virions in the stocks, TCID_50_ assays were performed in Vero E6 cells as described previously [59].

Particle densities of virions in PBS were determined at 25 °C by dynamic light scattering with the DynaPro NanoStar DLS instrument (Wyatt Technology, Santa Barbara, CA, USA). Calculations were performed using a spherical hydrodynamic model and integrating under the peak of hydrodynamic radius centered at 121 nm for EBOV-VSV and 104 nm for VSV.

### 4.5. In Vitro Selection of DNA Aptamers against sGP (Soluble Glycoprotein)

Systematic Evolution of Ligands by Exponential Enrichment (SELEX) was carried out to select DNA aptamers against EBOV sGP. The first round of SELEX involved incubating 10^15^ molecules (1.66 nmoles) of ssDNA SELEX library with 0.7 nmoles of sGP in PBS at 23 °C. The non-binding aptamers were removed by passing the sample through a nitrocellulose membrane (MiliporeSigma, Burlington, MA, USA, Cat. No. HAWP02500), to which the protein adsorbed with the bound DNAs. The protein-bound ssDNAs were extracted from the membrane by heating at 80 °C for 10 min in freshly prepared 7 M urea in water, then concentrated by ethanol precipitation. The extracted ssDNAs were amplified by PCR using 0.05 U/µL Taq DNA polymerase (GenScript # E00007), Taq Buffer (GenScript #B0005), 0.2 mM dNTPs (Genscript, Piscataway, NJ, USA, Cat. No. C01582-1), and 2 µM each of primer 485 and Biotin-5617. The quality of the PCR product was checked by resolution through 2% agarose gel in TAE buffer at a constant 110 V. The PCR products were purified using QIAquick PCR Purification Kit (Qiagen, Germantown, MD, USA, Cat. No. 28104) and quantified using a Nanodrop spectrophotometer ND1000 Spectrophotometer (Nanodrop Technologies). The biotinylated dsDNA was incubated with Dynabeads M280 (Dynal Biotech, Cat. No. 650.01) in BW for 20 min at 23 °C, followed by magnetic collection and a single wash with BW. The collected beads were incubated in 100 mM NaOH for 10 min, then the Dynabeads were removed, and the recovered DNA solution was neutralized with an equal volume of 100 mM HCl. The ssDNAs were concentrated by ethanol precipitation and used for the next round of SELEX. Successive rounds of selection included incubation with protein, capture on nitrocellulose, elution of ssDNA, PCR amplification, separation of the dsDNA by removing the biotinylated strand, and ethanol precipitation. A total of 8 positive rounds of selection were carried out, with a gradual increase in selection pressure by way of the ssDNA:sGP ratio (2:1, 3:1, 4:1, 5:1, 6:1, 7:1, 8:1, 10:1). A counter-selection was performed between the fourth and fifth rounds to remove nonspecific binders, such as oligonucleotides that bound nitrocellulose or oligonucleotides that bound HSA from the pool. The counter-selection involved incubating the 4th round of selected oligonucleotides at a 1:1 molar ratio with 3.7 µM HSA for 30 min at 23 °C, then passing this mixture through a nitrocellulose membrane. The enriched pool from the final SELEX round was cloned into the TOPO XL PCR cloning plasmid, and the resulting clones were sequenced.

### 4.6. End Labelling the ssDNA Oligonucleotides and In Vitro Translation of 2′FY-RNA for Characterizing Aptamers

DNA aptamers were end-labelled with ^32^P by incubating 1 µM DNA for 60 min at 37 °C with 0.2 µM ^32^P-ATP (7000 Ci/mmol) and 0.4 U/µL T4 polynucleotide Kinase (NEB, Ipswich MA, Cat. No. M0201L) in 70 mM Tris HCl, 10 mM MgCl_2_, and 5 mM dithiothreitol, at pH 7.6. The labelled oligonucleotides were purified using Biospin 6 (P-6 Gel) resins (Bio-Gel, Bio-Rad, Hercules, CA, USA, Cat. No. 1504130). The percentage of the ^32^P in each preparation that was incorporated into oligonucleotides was calculated from acid precipitable cpm determined by ascending thin layer chromatography (ATLC) using P30 Filtermat, WALLAC; a glass fiber filter with negatively charged P30 active groups (Perkin Elmer, Waltham MA, USA, Cat. No. 1450-523); and a mobile phase of 30% *v/v* methanol, 10% w/v TCA, and 10% *v/v* acetic acid. The ^32^P-labelled oligonucleotides were stored at 4 °C and refolded in PBS prior to use by heating at 95 °C for 5 min, followed by incubation at 23 °C for 30 min.

2′FY-RNA was prepared from dsDNA (5197 and 5198) by in vitro transcription (IVT) using a Durascribe^TM^ T7 transcription kit (Epicentre, Madison, WI, USA, Cat. #73369043). The resulting 2′FY-RNA was analyzed for concentration by UV/Vis spectroscopy and resolved through 8% polyacrylamide–7 M urea gel to confirm the size.

### 4.7. Single Well and Multiplex Filter Capture Assays

The DNA aptamers were incubated with the identified proteins at concentrations cited and in the sample buffers identified for individual experiments. The bound aptamers were captured by passing each aptamer–protein sample through a nitrocellulose membrane (MilliporeSigma, Cat. No. HAWP02500, Darmstadt, Germany) using a single-well binding apparatus or by capturing on a dot-blot apparatus (0.45 µ, 30 cm × 3.5 mm, BIO-RAD, #162-0115). Both assay types were performed under a constant vacuum. The membranes were washed each time with the sample buffer. The bound aptamers were quantified from the single membrane assays by liquid scintillation spectroscopy (Tricarb 4910TR, Perkin Elmer) and from the multiplex assay by exposing the membranes to phosphor screen, which was imaged by a Typhoon Imaging system (GE Healthcare, Waukesha, WI, USA, FLA9500). The images were quantified by ImageJ software [60].

### 4.8. Gel Electrophoresis, Western Blots, and ELISA

PAGE

Reducing and non-reducing SDS polyacrylamide gel electrophoresis (SDS-PAGE) for proteins were performed as previously described [61]. The gels were stained with Coomassie blue. Urea PAGE gels for oligonucleotides were 12% polyacrylamide with a running buffer of 7 M urea in TBE. The gels were stained with ethidium bromide.

Western blots

Irradiated serum obtained from an EBOV-infected macaque (kind gift of Dr. John Dye, USAMRIID) was serially diluted and separated by SDS PAGE [62] in parallel with a dilution series of purified sGP. Proteins were transferred to nitrocellulose and blocked with PS, 5% milk. The membrane was incubated with a pan filovirus GP monoclonal antibody (1:2000 21D10, IBT Bioservices) in PS-T for 1.5 h at room temperature and washed 4 times with PS-T. Donkey anti-mouse was conjugated to IRDye-800CW (1:5000) for 1.5 h at room temperature. The membrane was washed again 5 times with PS-T and imaged with a Li-Cor Odyssey. Pixel values of the sGP bands were obtained, and the standard curve was plotted to determine the quantity of sGP in the monkey serum.

ELISA

Purified sGP (50 ng/mL) was incubated overnight at 4 °C in 96-well Immulon 2 HB plates (ThermoFisher, Cat. No. 3455). Wells were blocked with 2% bovine serum albumin (BSA) for 1 h in PS. Plates were washed with PS, 0.15% Tween20, and incubated overnight at 4°C in PS, 2% BSA with antisera or mAbs, as noted in the figures and figure legends. Plates were washed four times with PS, 0.15% Tween20, at room temperature, and secondary antibodies (10 μg/mL) conjugated to HRP were added, and the samples were incubated at room temperature for one hour. Plates were washed five times with PS, 0.15% Tween20, and developed with TMB Substrate Reagent Set (SigmaAldrich, Cat. No. T4444, St. Louis, MI, USA) for 10 min. Development was stopped with 2 M H_2_SO_4_ and read at 450 nm by a microtiter plate reader.

Aptamer competition tested by ELISA

Purified sGP (50 ng/mL) was incubated overnight at 4 °C in wells of a 96-well Immulon 2 HB plates and then blocked with 2% BSA for 1 h in PBS. Plates were washed with PBS, 0.15% Tween20, and incubated with the identified oligonucleotides in PBS, 2% BSA, for 30 min at room temperature. Wells lacking oligonucleotides were in duplicate. Antibodies were incubated overnight at 4 °C (100 µL/well) at the following concentrations: α85-98 (1 to 100 dilution), 21D10 (0.4 μg/mL), FVM-04 (60 ng/mL), FVM-09 (7 ng/mL). The plates were washed four times with PS, 0.15% Tween20, and further developed and analyzed as for the ELISA assay described in the preceding section.

### 4.9. Aptamer and Protein Modeling and Docking

The EBOV sGP (PDB ID 5KEM) [18,63] and EBOV GP (PDB ID 5KEN) [18,64] were obtained from the RCSB Protein Data Bank site [65,66]. The Fab domains were removed from the structures. The secondary structures of aptamers were predicted using UNAfold webserver (http://www.unafold.org/ accessed on 6 October 2022) [66,67] with the following conditions: 137 mM Na^+^, 5 mM Mg^++^, 24 °C. The structures with the minimum free energy were chosen. The 3dRNA/DNA web server (http://biophy.hust.edu.cn/new/3dRNA accessed on 6 October 2022) [68,69] was used to predict the aptamer 3D structures (Appendix A). The HDOCK webserver (http://hdock.phys.hust.edu.cn accessed on 6 October 2022) [70,71] was used to dock the EBOV sGP to the aptamers. The top 10 docking results of each aptamer were chosen for further analysis. PyMOL 2.4.4 [72] was used for visualization and to determine the electrostatic potential. We used a PyMol script [73] to find the interface residues in the sGP dimer, which identified interface residues as those for which the difference in accessible surface areas calculated for the monomer and the dimer was more than 1.0 square angstrom. To compare the sequences of EBOV sGP (GenBank: AIO11752.1) and SUDV sGP (GenBank: ALT19780.1), these sequences were obtained from the National Center for Biotechnology Information (https://www.ncbi.nlm.nih.gov accessed on 6 October 2022) and aligned using protein blast (https://blast.ncbi.nlm.nih.gov/Blast.cgi accessed on 6 October 2022) [74].

### 4.10. Preparing the NAAO Membranes for Sensing Experiments

NAAO membranes (Sigma Aldrich, Whatman, Cat# WHA68096002) with nominal pore diameters of 20 nm and membrane thickness of 50 µm were used for the experiments. The NAAO membranes were sequentially sonicated in isopropanol, ethanol, and deionized distilled water (ddH_2_O) to clean the membrane surfaces. One side of the cleaned membranes was sputter-coated with 60 nm of gold, followed by washing with isopropanol, ethanol, and ddH_2_O. Note that 1 µM 5′thio-6011 (IDT, Coralville, IA, USA) in PS was stored in a refrigerator. Before an experiment, an aliquot of the aptamer in PS was heated to 90 °C, then brought to 5 mM MgCl_2_ and left for 40–60 min on the bench to slowly cool to room temperature before being placed on the gold-coated NAAO membrane, where it was left at 4 °C for 12 h to immobilize the aptamer. The NAAO membrane was washed with PBS and further incubated for 1 h at room temperature with 3 mM MCH in ddH_2_O to passivate the surface, followed by washing with PBS [25]. The sulfur groups of the MCH adsorb to the metal surface, forming an ordered and orientated monomolecular layer called a self-assembled monolayer (SAM). The aptamer concentration of 1 µM and immobilization at 150 mM NaCl were chosen to obtain grafting densities of 10^11^ to 10^12^ cm^−2^ on the gold surface. Grafting densities of 10^11^ and 10^12^ cm^−2^ are reported as desirable for good signal to noise (S/N) ratios. [75,76].

### 4.11. Electrochemical Sensing (EIS)

Details of the electrochemical sensing and the basis of its optimization have been described previously by us and others [25,77,78,79,80,81] and are schematically presented in Appendix A. A custom-performed Teflon cell was used for all electrochemical experiments. The four-electrode method was used to measure the impedance changes for the NAAO membrane. Platinum wires were used for the working (WE) and counter electrodes (CE), whereas Ag/AgCl wires (Invivometric) were used as the two reference electrodes (RE). EIS was carried out with an AC perturbation signal of 5 mV over a DC potential equal to the open circuit potential of the system, within the frequency range of 10 kHz–0.1 Hz. Equal volumes of protein solution (sGP as analyte and HSA for negative controls) prepared in PBS were injected onto the port of the Teflon cell containing the aptamer functionalized side of the membrane (Figure 7A) [81]. The concentration of each injection was determined to achieve the desired target concentration inside the electrochemical cell to obtain a calibration curve for the titration experiment. The LOD was calculated as:$$\mathrm{LOD}=3.3*\frac{\mathrm{standard}\;\mathrm{deviation}\;\mathrm{of}\;\mathrm{the}\;\mathrm{blank}\;\mathrm{signal}}{\mathrm{slope}\;\mathrm{in}\;\mathrm{the}\;\mathrm{linear}\;\mathrm{range}\;\mathrm{of}\;\mathrm{the}\;\mathrm{sensor}}$$

### 4.12. EIS of Infected Monkey Serum

Uninfected monkey serum (UMS) diluted to 10% in PBS was used as the electrolyte in the Teflon cell for the electrochemical testing. sGP mixed with UMS was used to create a calibration curve with several titration experiments using the procedure described above. The sensor responses for the calibration series were fit to the Langmuir equation below, for which B_min_ is the minimum binding value, B_max_ is the maximum binding value, and L is the ligand concentration:$$B=B_{\min}+\frac{B_{\max}L}{L+K_{d}}$$
The dissociation constant, K_d_ = 2.2 nM, and the regression constants, B_max_ = 1.01 and B_min_ = 0, obtained from the calibration curve, were used to predict the sGP concentration in the monkey serum sample based on the sensor responses for the sample.

To obtain sensor responses for the monkey serum, equal volumes of infected monkey serum (IMS) prepared by diluting in 10% uninfected monkey serum (UMS) were injected into the port of the electrochemical cell containing the aptamer-functionalized side of the membrane. The electrochemical impedance response was modeled using a circuit consisting of two parallel resistors and capacitor loops in series [35,82]. It was observed that the membrane resistance varied strongly with the introduction of the IMS. The sensor response was calculated as the change in membrane resistance (ΔR), after injection of the IMS with the background signal subtracted that was obtained after injection of UMS; S = ΔR_IMS_ − ΔR_UMS_. This sensor response “S” was normalized with maximum response (S/S_max_), which was then fit to the Langmuir equation (above) to predict the concentration of sGP present in the IMS based on a calibration curve for purified sGP in UMS (Figure 7C and Appendix A). Appendix A lists other reported sensors that detect *Ebolavirus* [83,84,85,86,87,88,89,90,91].

### 4.13. Analysis of Binding Isotherms and Statistical Evaluations

All binding isotherms were fit to the Langmuir equation (above), and Sigmaplot was used to obtain statistical parameters. All reported values for K_d_ passed the normality (Shapiro–Wilk) and the constant variance (Spearman rank correlation) tests. Standard deviations, shown as error bars in the graphs, were calculated to include the errors in the average values in the blanks by the formula: $Standard\;deviation=\sqrt{{\mathrm{Se}}^{2}+{\mathrm{Sb}}^{2}}$, where Se = standard deviation of the sample and Sb = standard deviation of the subtracted blank.

## 5. Patents

S.B. and M.N-H are inventors of U.S. patent 11231420, Selection and Optimization of Aptamers to Recognize Ebola Markers.

## Figures and Tables

**Figure 1 ijms-24-04627-f001:**
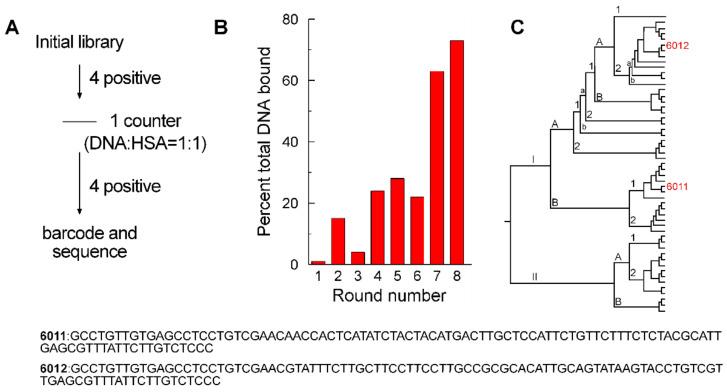
**SELEX experiment results:** (**A**) Single-stranded DNA aptamers were obtained by multiple rounds of positive selection against sGP and one counter-selection against the filter capture matrix and HSA. (**B**) The percentage of each successive pool that was captured with EBOV sGP in each round increased with subsequent rounds. (**C**) The results of topo-cloning are presented as a cladogram showing the positions of the sequences for 6011 and 6012.

**Figure 2 ijms-24-04627-f002:**
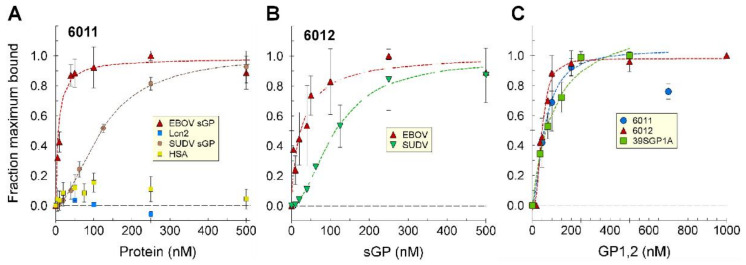
**Binding isotherms for oligonucleotides 6011, 6012 and 39SGP1A**. (**A**) Binding isotherms for 6011 with EBOV sGP, SUDV sGP, His-Lcn2 or HSA at the concentrations identified by the x-axis labels. The numbers of independently performed experiments, each assessing duplicate or triplicate samples, that contributed to these binding isotherms, were EBOV sGP (7), SUDV sGP (2), Lcn2, and HSA (1). (**B**) Binding isotherms for 6012 with EBOV sGP and SUDV sGP at the concentrations identified by the x-axis labels. The numbers of independently performed experiments, each assessing duplicate or triplicate samples, that contributed to these binding isotherms, were EBOV sGP (5) and SUDV sGP (2). (**C**) Binding isotherms of 6011 and 6012 for EBOV GP1,2 at the concentrations identified by the x-axis labels. The numbers of independently performed experiments, each assessing triplicate samples (with the exception of 6011, which was performed in duplicate), which contributed to these binding isotherms, were 6011 (3), 6012 (2), and 39SGP1A (2). In each titration, the concentrations of the proteins are for the sGP as homodimer and GP1,2 as monomer. All binding assays were performed in PBS at 24 °C. Standard deviations are shown.

**Figure 3 ijms-24-04627-f003:**
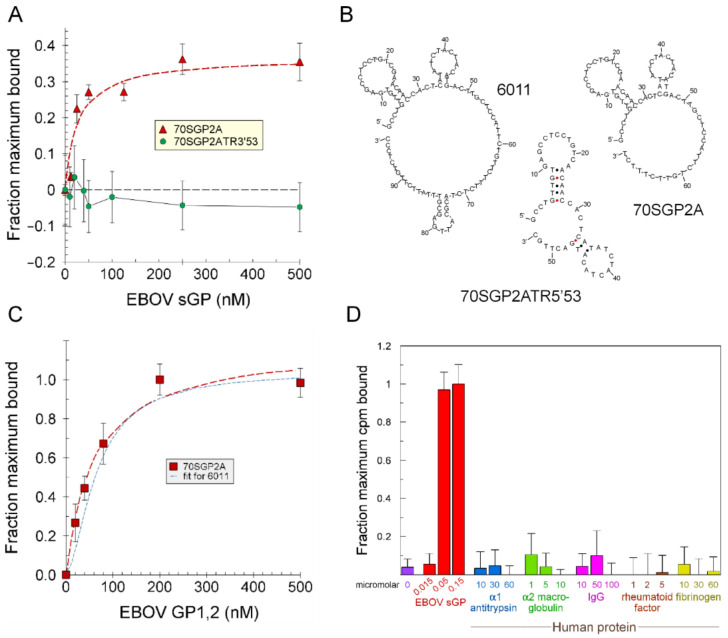
**Activities of 6011 truncations**. (**A**) Binding of 70SGP2A and 70SGP2ATR5′53 to EBOV sGP at the concentrations identified by the x-axis labels. (**B**) Predicted 2D structures of 6011, 70SGP2A, and 70SGP2ATR5′53, (**C**) Binding of 70SGP2A to EBOV GP1,2. The blue dotted line is the fit line from Figure 2C for the binding isotherm of 6011 with GP1,2. (**D**) The specificity of 70SGP2A for sGP and GP1,2 was tested with ^32^P-70SGP2A incubated with the identified concentrations of each protein. The amount of ^32^P-70SGP2A bound to protein was determined by filter capture for each condition. Cpm bound for duplicates for each condition were averaged, and the averages of blanks (no protein) were subtracted from values of samples with protein. The average cpm (-blank) bound for each condition was normalized to the average cpm (-blank) bound by 0.15 µM sGP. In each titration, the concentrations of the proteins are for the quaternary structures, which are the sGP homodimer, the IgG heterotetramer, and monomers for the other listed proteins. All binding assays were performed in PBS at 23 °C. Standard deviations were calculated as described in materials and methods to include the error in the blank when that was subtracted.

**Figure 4 ijms-24-04627-f004:**
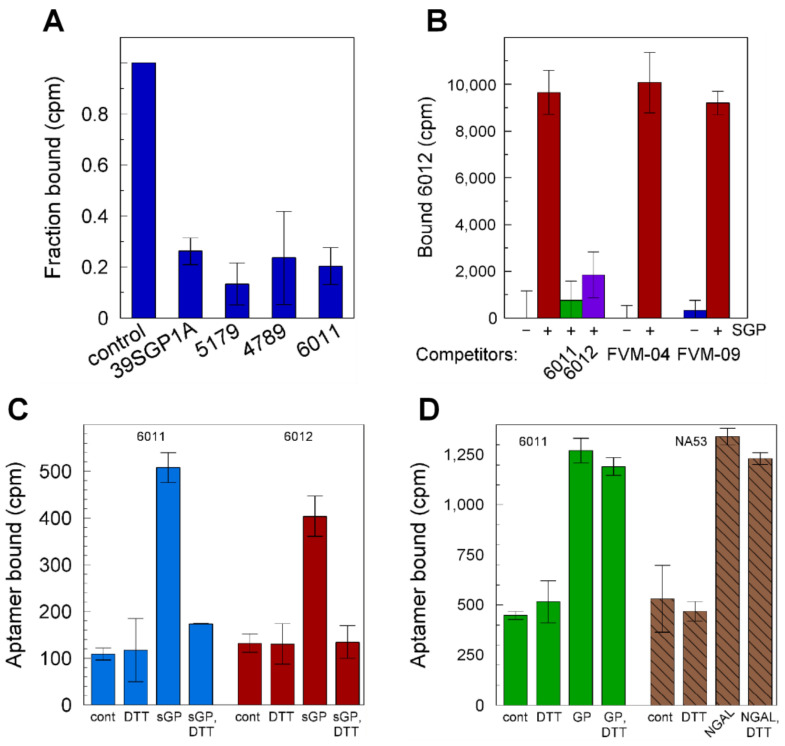
**Analysis of aptamer and antibody binding to EBOV sGP**. (**A**) Binding of 200 nM ^32^P-39SGP1A to 4 µM EBOV sGP in the presence of the listed oligonucleotides, each at 13.3 µM. (**B**) Binding of 10 nM ^32^P-6012 to EBOV sGP in the presence or absence of 6011 or 6012 (each at 102 nM) or in the presence or absence of antibody, FVM-04 or FVM-09 (each at 1 µM). (**C**) Binding of 10 nM ^32^P-6011 (blue) or ^32^P-6012 (red) (each at 10 nM) to 1 µM EBOV sGP in the presence or absence of 10 mM DTT. (**D**) The effect of 10 mM DTT on the binding of 10 nM ^32^P-6011 to 0.5 µM EBOV GP1,2 (green) or ^32^P-NA53 to NGAL (hatched brown). Binding assays were performed in PBS (6011, 6012) or NBB (NA53) at 23 °C. Standard deviations were calculated, as described in Materials and Methods, to include the error in the blank when that was subtracted.

**Figure 5 ijms-24-04627-f005:**
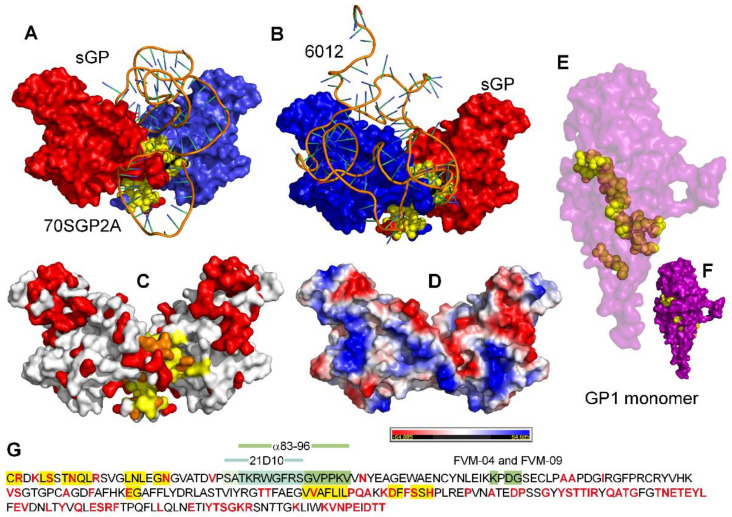
**Models that predict the interaction of sGP with DNA aptamers.** EBOV sGP (PDB ID 5KEM) showing chain A (red), chain F (blue), and the interface residues of the EBOV sGP dimer (yellow) docked with the predicted 3D structures of (**A**) of 70SGP2A and (**B**) oligonucleotide 6012. (**C**) A comparison of the 3D structures of EBOV sGP (PDB ID 5KEM) and SUDV sGP with conserved residues (white), divergent residues (red), and the interface residues conserved in the EBOV and SUDV sGP dimer as yellow and divergent interface residues as orange. (**D**) The electrostatic potential surface of EBOV sGP (PDB 5KEM). Blue residues are positively charged, red residues are negatively charged, and white residues are neutral. (**E**) All interface residues of the EBOV sGP dimer residues, including surface exposed (yellow) and buried (brown) in the GP1,2 monomer (PDB ID 5KEN). (**F**) Only the interface residues of the EBOV sGP dimer residues that are exposed on the GP1,2 monomer surface (yellow). (**G**) The primary sequence of mature EBOV sGP with divergent residues in red and the interface residues highlighted in yellow. The known antibody epitopes are identified by green highlighting, and their positions on the 3D structure are shown in Appendix A.

**Figure 6 ijms-24-04627-f006:**
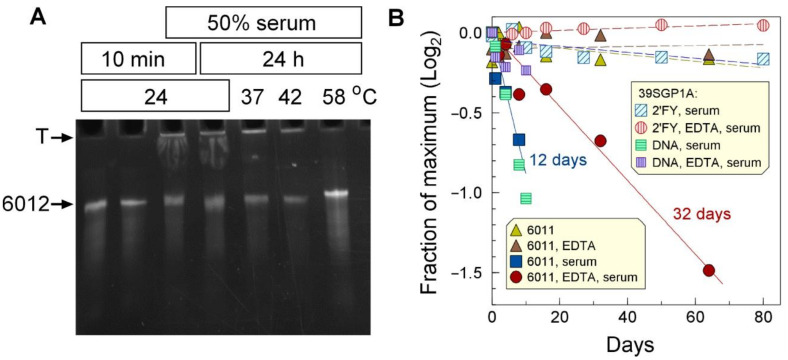
**Stability of aptamers in relation to temperature and serum components**. (**A**) Twenty-micromolar oligonucleotide 6012 was incubated in PBS at the temperatures and for the times shown in the figure, with or without 50% human serum. (**B**) Thirty-micromolar 6011 was incubated at 24 °C for the identified times in BSM1C with and without 2 mM EDTA (gold and brown triangles). Sixteen-micromolar 6011 was incubated at 24 °C in PS, 10mM NaN_3_, and 50% bovine serum in the presence or absence of 1 mM EDTA (blue and red-filled symbols). Forty-micromolar 39SGP1A was incubated at 24 °C for the identified times in RSBA and 50% bovine serum, in the presence or absence of 1 mM EDTA (vertically hatched red circles, diagonally hatched blue squares, respectively). Forty-micromolar 39SGP1A as DNA was incubated at 24 °C for the identified times in RSBA and 50% bovine serum, in the presence or absence of 1 mM EDTA (vertically hatched mauve squares, horizontally hatched green squares, respectively).

**Figure 7 ijms-24-04627-f007:**
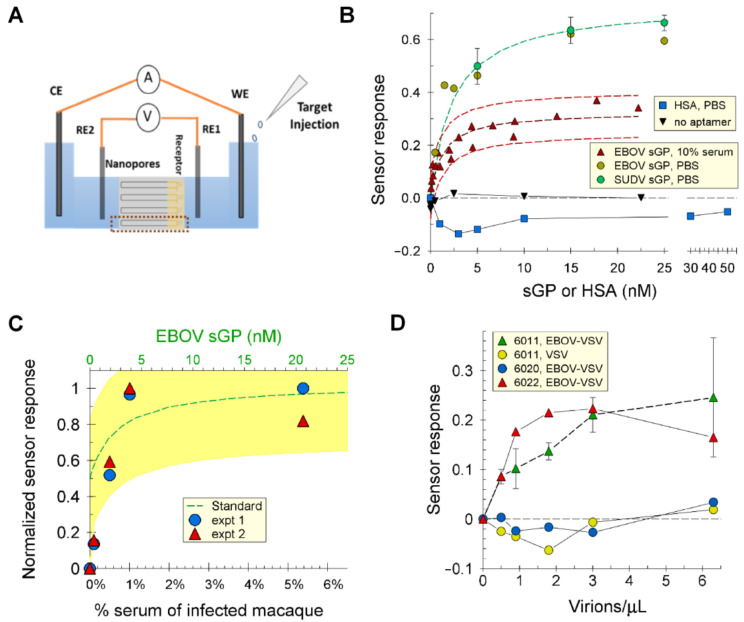
**Electrochemical sensing**. (**A**) Sensing cell setup. (**B**) When mounted on an NAAO membrane, oligonucleotide 6011 produced an electrochemical signal in response to EBOV sGP in the presence of 10% human serum (red triangles) and in its absence (gold circles). It also responded to SUDV sGP (green circles) with the same binding isotherm as for EBOV sGP. The aptasensor did not respond to the addition of HSA (blue squares). There was no electrochemical response from the NAAO membrane lacking the aptamer (inverted triangles). The figure shows the results of single titrations for no aptamer, has, and EBOV sGP in PBS; an average of duplicate independent titrations for SUDV sGP; and a compiled set of data from 3 independent titrations for EBOV sGP in 10% human serum. Fit lines are shown for SUDV sGP in PBS (dashed green) and EBOV sGP in 10% serum (dashed dark red) with red dashed lines above and below the fitline to show the range of average deviation (± 5%) from the fit for EBOV sGP in 10% serum. (**C**) The NAAO sensor conjugated with oligonucleotide 6011 was used to titrate infected macaque serum in a background of 10% macaque serum. The results of two experiments are shown. A standard curve with recombinant sGP in 10% uninfected macaque serum (green dashed line) was used to determine a concentration of 1 to 1.3 µM in the infected serum. The yellow highlighted region identifies the 95% confidence level for the standard curve. (**D**) EBOV pseudotyped virions (EBOV-VSV) or VSV without glycoprotein (VSV) in PBS were titrated in the electrochemical sensor functionalized with 5′thio-oligonucleotides, 6011, 6020, and 6022. The results show the average of 2 independent experiments for 6011 with EBOV-VSV, 3 independent experiments for 6011 with VSV, and 1 experiment each for 6020 and 6022. Shown are the average values with the standard deviations. The concentrations of sGP are based on the dimer molecular weight. The density of virions was determined by dynamic light scattering. The sensor response (S) was calculated as S = ΔR/R, and the normalized sensor response was S/S_max_. Nyquist plots for these data can be found in Appendix A.

**Table 1 ijms-24-04627-t001:** Binding affinities of three aptamers.

Aptamer	Nucleic Acid Type	sGP, EBOV (K_d_, nM)	sGP, SUDV (K_d_, nM)	GP1,2, EBOV (K_d_, nM)
6011	DNA	8.5 ± 3.2	165 ± 44	48 ± 32
6012	DNA	27 ± 12	151 ± 44	54 ± 23
39SGP1A	2′FY-RNA	* 13 ± 5	* 147 ± 59	104 ± 31
70SGP2A	DNA	27 ± 15	ND	58 ± 17

Affinities were determined using a filter binding assay. The K_d_ and error in the estimate were determined using Sigmaplot. * Data used to calculate K_d_s for 39SGP1A from [11]. Note that the K_d_s were determined with the assumption that one aptamer binds one protein molecule. For this analysis, one protein molecule is a dimer of sGP and a monomer of GP1,2. ND, not determined.

**Table 2 ijms-24-04627-t002:** Oligonucleotides used in this study.

Oligonucleotide	Nucleic Acid Type	Sequence
SELEX library	DNA	GCCTGTTGTGAGCCTCCTGTCGAA (53N) TTGAGCGTTTATTCTTGTCTCCC
6011	DNA	GCCTGTTGTGAGCCTCCTGTCGAACAACCACTCATATCTACTACATGACTTGCTCCATTCTGTTCTTTCTCTACGCATTGAGCGTTTATTCTTGTCTCCC
6012	DNA	GCCTGTTGTGAGCCTCCTGTCGAACGTATTTCTTGCTTCCTTCCTTGCCGCGCACATTGCAGTATAAGTACCTGTCGTTGAGCGTTTATTCTTGTCTCCC
39SGP1A [11]	2′FY-RNA	GGGCGCUCAAUUUUUUAUUGCAUUUUUCUUUGAGCGCCC
70SGP2A	DNA	GCCTGTTGTGAGCCTCCTGTCGAACAACCACTCATATCTACTACATGACTTGCTCCATTCTGTTCTTTCT
6020	DNA	GCCTGTTGTGAGCCTCCTGTCGAACATACCGTTCCACCCACATTTCAACCTTCATCCATCCTATTATTAGCCCACTCTTGAGCGTTTATTCTTGTCTCCC
6022	DNA	GCCTGTTGTGAGCCTCCTGTCGAACCCTATCTTGTTCATGCTATTCTCAATATTTTCGGTTCACTTACCGTCTGCCTTTGAGCGTTTATTCTTGTCTCCC
485	5′ DNA primer for ssDNA library	GCCTGTTGTGAGCCTCCTGTCGAA
5617	3′ DNA primer for ssDNA library	GGGAGACAAGAATAAACGCTC (this primer was biotinylated for SELEX)
5197	DNA template for IVT	TAATACGACTCACTATAGGGCGCTCAATTTTTTATTGCATTTTTCTTTGAGCGCCC
5198	DNA complement for IVT	GGGCGCTCAAAGAAAAATGCAATAA AAAATTGAGCGCCCTATAGTGAGTCGTATTA
NA53 [24]	DNA	AGCAGCACAGAGGTCAGATGCCGTGCGGATGTACAGGGACTTGGATAGTTTCTGACCTATGCGTGCTACCGTGAA

## Data Availability

Data is available on reasonable request to marit@iastate.edu.

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
