# Peer review of "Structurally Different Yet Functionally Similar: Aptamers Specific for the Ebola Virus Soluble Glycoprotein and GP1,2 and Their Application in Electrochemical Sensing"

_ijms, 2023, doi:10.3390/ijms24054627_

Round 1

Reviewer 1 Report

In this manuscript, the authors suggested an aptamer for Ebola virus soluble glycoprotein and examined their application in electrochemical sensing. The electrochemical part of the manuscript needs major revision.

1-In Fig 7B. The author claimed that they have a linear response in the range of 0.5 -2 nM. This linear range is not clear in Fig.7B, the author needs to show exactly the linear range part in a separate plot with all the EIS curves.

2-In Fig.7C, the author has to explain more about the normalization of the response. More discussion About the mentioned equations on page 10, line 324, and how they drew the Fig7C is required.

3-The author needs to calculate the association constant (Ka) of the aptamer -target with the electrochemical methods

4-In the electrochemical sensing part, the author needs to explain how they immobilized aptamer on the sensing surface, and please explain the optimization steps for the aptamer concentration. how long is the immobilization time? The incubation time of aptamer with the target? And other optimization steps.

5-It needs to be explained the reason for using the NAAO membrane.

6-The author needs to include all the EIS curves for the selectivity assay.

7-Please prepare a table comparing the advantages and disadvantages of the proposed aptamer with other aptamers introduced in scientific reports for the Ebola virus

8-The author needs to provide one scheme for illustrating the steps of preparation for the NAAO membrane and electrochemical sensor

9- In the introduction, the author needs to explain the important details of the electrochemical sensing procedure.

Author Response

We thank the reviewer for the excellent observations and suggestions and have revised the manuscript accordingly. Below is our point by point response with the reviewer's comments preceding our response, which is italicized:

1-In Fig 7B. The author claimed that they have a linear response in the range of 0.5 -2 nM. This linear range is not clear in Fig.7B, the author needs to show exactly the linear range part in a separate plot with all the EIS curves.

The data showing the linear range is now in Figure S9A. We have also corrected the rounding error for the range of linearity, which was 0.08 to 0.8 nM sGP and simplified to fit to a Langmuir fit as the n value is clearly no larger than 1.

2-In Fig.7C, the author has to explain more about the normalization of the response. More discussion About the mentioned equations on page 10, line 324, and how they drew the Fig7C is required.

We have expanded the discussion of the method (4.12) to describe the process and included the calibration curve as Figure S9A.

 3-The author needs to calculate the association constant (Ka) of the aptamer -target with the electrochemical methods

To be consistent with other reports of affinity in the manuscript, the affinity of the electrochemical sensor reported as Kd  (2.2 nM), which is the inverse of the Ka.

4-In the electrochemical sensing part, the author needs to explain how they immobilized aptamer on the sensing surface, and please explain the optimization steps for the aptamer concentration. how long is the immobilization time? The incubation time of aptamer with the target? And other optimization steps.

This information is found in section 4.10. Preparing the NAAO membranes for sensing experiments. We have also included information regarding the choice of optimal conditions for SAM coverage.

5-It needs to be explained the reason for using the NAAO membrane.

A section has been added to the discussion regarding the advantages of the NAAO membranes for sensing.

6-The author needs to include all the EIS curves for the selectivity assay.

These have been provided in Figure S9.

7-Please prepare a table comparing the advantages and disadvantages of the proposed aptamer with other aptamers introduced in scientific reports for the Ebola virus

 This is provided as Table S1

8-The author needs to provide one scheme for illustrating the steps of preparation for the NAAO membrane and electrochemical sensor

We have cited the reference (Gosai et al., 2019) in the Materials and Methods section where we have published the requested scheme and have inserted the scheme with permission from the publisher as Fig. S10 of the Supplemental material.

9- In the introduction, the author needs to explain the important details of the electrochemical sensing procedure.

We have included a paragraph on impedance-based sensing devices in the introduction.

Reviewer 2 Report

In this work, S. Banerjee et al. have performed a SELEX for the Soluble glycoprotein of the Ebola virus, obtaining two aptamers with good affinity and selectivity towards the protein. They studied the binding site of the aptamers to the protein with empirical studies and with computational modelling of the complex. Furthermore, they studied the stability of the aptamers in serum and their application in the development of an electrochemical sensor for the detection of the protein in serum. Publication in the International Journal of Molecular Sciences could be considered after addressing the minor issues detailed in the comments below:

1.     The introduction is short. I suggest the addition of some background information about other aptamers described in the literature for the same protein, and about the available methods for the detection of the protein. Which is the standard method used in clinical practice?

2.      I would like to know how the authors selected the two aptamers from the family tree with the 53 clones.

3.     In table 1, the authors should include the units of the KD.

4.     It is difficult to read the legend of the figures. The authors should improve the quality of the figures.

5.     In the electrochemical biosensors section, I think it would be helpful to incorporate a table that compares the results of your work with the results of others' works published in journals.

Author Response

We thank the reviewer for the excellent observations and suggestions and have revised the manuscript accordingly. Below is our point by point response with the reviewer's comments preceding our response, which is italicized:

  1. The introduction is short. I suggest the addition of some background information about other aptamers described in the literature for the same protein, and about the available methods for the detection of the protein. Which is the standard method used in clinical practice?

We have expanded the introduction to discuss current means of detecting Ebolavirus and the consequence of timeliness to its spread.

  1. I would like to know how the authors selected the two aptamers from the family tree with the 53 clones.

Oligonucleotides from the major clades were tested for binding by the dot-blot assay. We have updated the description with this information.

 In table 1, the authors should include the units of the KD.

This has been corrected. We apologize for the omission.

 It is difficult to read the legend of the figures. The authors should improve the quality of the figures.

 The text size and background color have been changed in the figures to increase readability. All figures with legends in the figure have been updated in this way.

  1. In the electrochemical biosensors section, I think it would be helpful to incorporate a table that compares the results of your work with the results of others' works published in journals.

We thank the reviewer for this suggestion and have added Table S2 (Comparison of sensors for Ebola virus).

Round 2

Reviewer 1 Report

This manuscript is not acceptable based on the response provided by the author:

1. As we can see, in Fig.S9 the response of the sensor is not in a linear range, especially since the result of 6.4 nM of sGP is very far from the linear line. Also, the author needs to explain the reason for having two semicircles in Nyquist plots, and how they calculated Rct when they have seen two semicircles! Also, why some sensors are showing two semicircles in Nyquist plots and some of them one semicircle?

2. Manuscript contains several misprints. Language style should be substantially improved. 

3. We asked the authors to provide optimization steps for aptamer immobilization and incubation time but the explanation in section 4.10 is not related to the optimization steps.

Author Response

Please see attached response letter

Round 3

Reviewer 1 Report

This manuscript is not acceptable based on the response provided by the author, and need minor revision:

- Sorry for not mentioning the exact concentration, if you look at the following plot (Fig.S9A) which the author claimed showing the linear response of the sensor and Nyquist plots for data contributing to the plot, the response between 0.2-0.3nM is out of linear range. How can we claim a linear response?

Author Response

We understand the reviewer’s concerns regarding the question of range of linearity. First, we want to stress that our analysis is through curve fitting using the Langmuir equation and not by using a linear transformation. We also note that the error range for each standard curve is shown in Figure 7 by the dashed lines in Fig. 7B and the yellow band in figure 7C. The variations that the reviewer noted in the plot in figure S9A, which covered the lower (semi-linear) portion of the Langmuir plot, are within these error estimates. However, in response to the reviewer’s request, we have decided to present a linear transformation of the larger set of data that was used to determine the Kd for the sensor (Fig. 7B) as is now shown in Figure S9 (A,B). We also show the data for the standard curve for the Macaque monkey serum (Fig. S9C), which gave a similar binding isotherm and Kd estimate. For convenience and comparison between experiments, these data were normalized to the maximum value in each dataset. As the reviewer can see, the data is linear over the range of 0.15 to 44 nM (Fig. S9B).

We appreciate the reviewer’s persistence with respect to the issue of linearity as we realized that the values for the range reported in the manuscript were incorrect, corresponding only to the initial portion of Langmuir isotherm and not to the full linear range in a semi-log transformation. We have corrected the manuscript for the linear range (line 329).

With our current revisions and additional representation of the data, we hope that the manuscript is now ready for publication.

Yours sincerely,

Marit Nilsen-Hamilton